

# Enhanced diapycnal mixing by polarity-reversing internal solitary
# waves in the South China Sea
**Yi Gong[1], Haibin Song[1]\*, Zhongxiang Zhao[2], Yongxian Guan[3], Kun Zhang[1], Yunyan Kuang[1],**
**Wenhao Fan[1]**
1 State Key laboratory of Marine Geology, School of Ocean and Earth Science, Tongji University, Shanghai 200092,
China
2 Applied Physics Laboratory, University of Washington, Seattle, WA, USA
3 MNR Key Laboratory of Marine Mineral Resources, Guangzhou Marine Geological Survey, China Geological
Survey, Guangzhou 510760, China
[*]Corresponding author. hbsong@tongji.edu.cn
## Abstract
Shoaling internal solitary waves near the Dongsha Atoll in the South China Sea dissipate their
energy and thus enhance diapycnal mixing, which have an important impact on the oceanic
environment and primary productivity. The enhanced diapycnal mixing is patchy and instantaneous.
Evaluating its spatiotemporal distribution requires comprehensive observation data. Fortunately,
seismic oceanography meets the requirements, thanks to its high spatial resolution and large spatial
range. In this paper, we studied three internal solitary waves in reversing polarity near the Dongsha
Atoll, and calculated the spatial distribution of resultant diapycnal diffusivity. Our results show that
the average diffusivities along three survey lines are two orders of magnitude larger than the open-
ocean value. The average diffusivity in the internal solitary wave with reversing polarity is three
times that of the non-polarity-reversal region. The diapycnal diffusivity is higher at the front of one
internal solitary wave, and gradually decreases from shallow to deep water in the vertical direction.
Our results also indicates that (1) the enhanced diapycnal diffusivity is related to reflection seismic
events; (2) convective instability and shear instability may both contribute to the enhanced diapycnal
mixing in the polarity-reversing process; and (3) the difference between our and previous diffusivity
profiles is about 2-3 orders of magnitude, but their vertical distribution is almost the same.
[Key words] Internal solitary waves, Polarity reversal, Diapycnal mixing, Northeastern South China
Sea, Seismic oceanography.

## 1. Introduction
Energy dissipation of internal waves enhances diapycnal mixing, and turbulence in the form of
internal wave breaking is the primary mechanism for mixing thermodynamic properties in the ocean
(St. Laurent et al., 2011). Small-scale changes of topography also significantly enhances local
mixing (Nash and Moum, 2001; Klymak et al., 2008; Palmer et al., 2013; Staalstrøm et al., 2015;
Wijesekera et al., 2020; Voet et al., 2020). Internal tides and internal waves are ubiquitous on the
global continental shelves and slopes (Holloway et al., 2001). They play an important role in the
global oceanic energy balance and provide energy for ocean mixing (Mackinnon and Gregg, 2003).
Due to shoaling internal waves and seafloor roughness, turbulent mixing on the continental shelves





and slopes is more variable than in the open ocean (Carter et al., 2005). Diapycnal diffusivity
observed on continental shelf and slope can span four orders of magnitude (Gregg and Özsoy, 1999;
Nash and Moum, 2001). Internal solitary wave is a kind of nonlinear internal wave, which usually
carries a large amount of energy. Numerical simulation results indicate that up to 73% of the internal
wave field energy may be in internal solitary waves (Bogucki et al., 1997). Therefore, internal
solitary waves propagating to the continental shelf and slope can greatly change the local mixing. A
number of researches have been carried out on mixing caused by internal solitary waves on the
continental shelf and slope. Observations have shown that the turbulence induced by shear
instability at the rear of internal solitary wave sharply increases the mixing (Sandstrom et al., 1989;
Sandstrom and Oakey, 1995; Moum et al., 2003; Richards et al., 2013). Mackinnon and Gregg (2003)
estimated that 50% of the dissipation in the thermocline occurred during the propagation of internal
solitary waves according to their observations. The elevation internal solitary waves propagating
near the seafloor enhances mixing, resuspending and transporting materials, which has an important
impact on the local ecological environment (Klymak and Moum, 2003; Moum et al., 2007).
Internal solitary waves are ubiquitous in the northeastern South China Sea (Zhao et al., 2003;
Klymak et al., 2006; Xu et al., 2010; Cai et al., 2012; Alford et al., 2015). They are generated by
interaction of internal tides and topography in the Luzon Strait and propagate toward Dongsha Atoll,
where their energy is dissipated in shoaling. The continental shelf and slope of the northeastern
South China Sea is close to the source, so that the amplitude and energy of internal solitary waves
in this area are large. The energy dissipation of internal solitary waves occurs most near Dongsha
Atoll and its southeastern shelf (Lien et al., 2005; Chang et al., 2006; St. Laurent, 2008).
Observations show that high turbulence mainly occurs in the continental shelf region, and the
average diffusivity can reach the order of O(-3) m2 s-1, while the diffusivity in the continental slope
region is one order of magnitude lower (Yang et al., 2014). When nonlinear internal waves travel
cross the continental slope, the waveform changes into different types (Terletska et al., 2020). In
this process, mixing is enhanced, and about 30% of the energy dissipation occurs near the seafloor
(St. Laurent, 2008). The energy flux of internal solitary waves around the Dongsha Plateau is large.
Lien et al. (2005) estimated that, if all nonlinear internal waves break within water depth of 10 m
and in a range of $200 \times 200$ km$^2$ centered on Dongsha Plateau, the magnitude of diffusivity can
exceed an order of O( -3). In addition, internal solitary waves shoaling near the Dongsha Atoll also
dissipate a lot of energy and improve the local mixing efficiency (Orr and Mignerey, 2003; St.
Laurent et al., 2011). The water in the northeastern South China Sea can exchange heat with the
water in the Pacific Ocean through the Kuroshio, and heat can be transferred to atmosphere through
the sea-air interface on the continental shelf as well. Therefore, internal solitary waves are an
important link for energy transfer in the South China Sea and play an important role in our
understanding of energy transfer between the ocean and climate environment.
Turbulence in the ocean is patchy and instantaneous. Therefore, it requires extensive observations
to accurately evaluate turbulent mixing (Whalen et al., 2012; Waterhouse et al., 2014; Kunze, 2017).
Seismic oceanography (Holbrook et al., 2003) has the advantages of wide observation range and
high spatial resolution (Ruddick et al., 2009), which is suitable for observing the spatial distribution
of turbulent mixing. Sheen et al. (2009) used reflection seismic data to give a diffusivity section of
oceanic front in the South Atlantic. Holbrook et al. (2013) comprehensively introduced the



theoretical basis for evaluating turbulent mixing from reflection seismic data. Subsequently, a large
number of scholars have used the reflection seismic method to study the spatial distribution of
turbulent mixing in different ocean regions or turbulent mixing induced by different ocean
phenomena (Fortin et al., 2016; Sallares et al., 2016; Dickinson et al., 2017; Mojica et al., 2018).
In this article, we used two-dimensional seismic data to observe the propagation of internal solitary
waves near the Dongsha Atoll, and calculated the spatial distribution of local diapycnal diffusivity
to evaluate the impact of internal solitary wave shoaling on turbulent mixing. Section 2 introduces
seismic data processing and the method of calculating turbulence mixing parameters. Section 3
describes the polarity reversal of internal solitary waves, horizontal slope spectrum and distribution
of turbulence diffusivity. In section 4 we analyze the relationship between diapycnal diffusivity and
reflection seismic events, and discuss the mechanism of turbulent mixing induced by internal
solitary waves. Besides, we compare the mixing scheme with our results. Section 5 gives a summary.

## 2. Data and methods
### 2.1. Seismic data acquisition and processing

The water is shallow on the continental shelf and slope near the Dongsha Atoll, so internal solitary
waves reach the transition point and their polarity changes from depression to elevation. In the
summer of 2009, the Guangzhou Marine Geological Survey (GMGS) set up a two-dimensional
seismic observation network in the Dongsha area. We found three internal solitary waves during the
polarity reversal process on the L1, L2, and L3 survey lines of the seismic data. The survey lines
are shown in Figure 1a, b. The streamer used in the acquisition process has a total length of 6 km
and 480 channels, the trace interval is 12.5 m, and the sampling interval is 2 ms. The airgun source
capacity is 5080 in$^3$ (1 in=2.54 cm), and the main frequency of the source is 35 Hz. The shot interval
is 25 m, and the minimum offset is 250 m. The time interval of shots is about 10 s. Survey lines L1
and L2 are the in-lines, which were from the southeast to the northwest. Survey line L3 is a cross-
line, which was from the southwest to the northeast. We calculated the mean buoyancy frequency
(Figure 1c) of the region around seismic survey lines (latitude range 21.5°-22.5°, longitude range
116°-118°, blue box in Figure 1a) by reanalysis temperature and salinity data with a water depth of
100-350 m. This depth range matches the observation depth of the seismic data. The hydrographic
data are provided by Copernicus Marine Environment Monitoring Service (CMEMS).

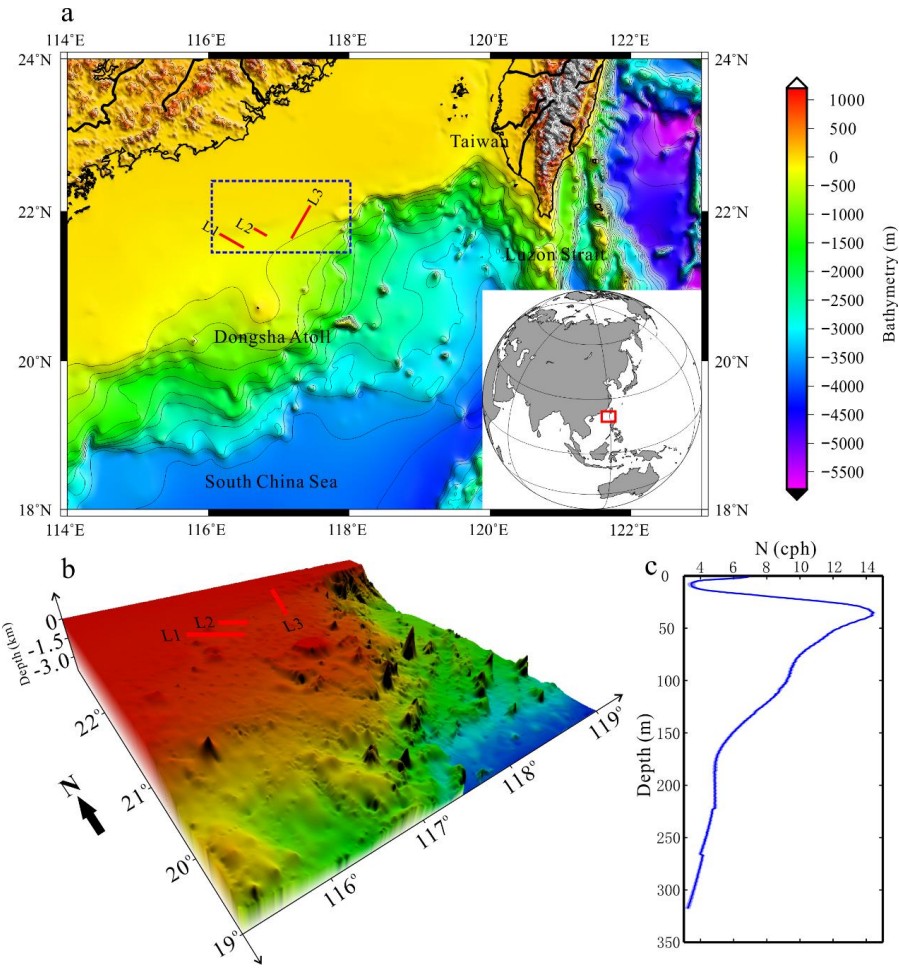


Figure 1. Bathymetry of the Dongsha area and the locations of seismic survey lines. (a) 2D bathymetry

map of the northeastern of South China Sea, with the red lines representing the seismic survey lines. (b)

3D bathymetry map around the Dongsha Atoll. (c) The mean buoyancy frequency around seismic survey

lines (blue box in (a)) and its 95% confidence interval (blue shadow).

After a conventional processing of the seismic data, an image of the ocean interior's structure can

be obtained. This image can be approximated as a temperature or salinity gradient map of the water

column (Ruddick, et al., 2009). The conventional processing of seismic data has 5 main steps,

including defining the observation system, noise and direct wave attenuation, velocity analysis,

normal moveout (NMO) and horizontal stacking. Then we use a bandpass filter to filter out low-

frequency noise below 8 Hz and high-frequency noise above 80 Hz. According to the linear

characteristics of the direct wave, we use a median filter to extract the direct wave signal, and

subtract it from the original signal to achieve the purpose of attenuating the direct wave.

Subsequently, we sorted the seismic data from shot gathers into common midpoint gathers (CMPs).

Sound speed is a function of depth and obtained through velocity analysis, and then the NMO is


applied to CMPs according to the function to flatten the reflection seismic events of the water
column. When NMO is applied, the seismic wave with large offset will be stretched, and the
stretched seismic waves need to be cut off. Usually, the default method is to use a linear function to
remove the stretched seismic waves (Figure 2a). This may lose a lot of shallow reflection signals
(Figure 2b). Bai et al. (2017) used the common offset seismic section to supplement the missing
information in shallow water, but the low signal-to-noise ratio of the common offset seismic section
cannot guarantee the imaging quality. In order to retain more shallow reflection signal, we used a
custom function to cut off the NMO stretch (Figure 2c), thereby satisfying the imaging requirement
of the shallow water column (Figure 2d). Finally, the seismic section of the water column can be
obtained by stacking the processed CMPs. Due to the shallow water depth, the seismic data is
seriously affected by swell noise. We filtered out the components of stacked seismic data in wave
number range corresponding to swells in the frequency-wave number domain. A detailed description
of the seismic data processing can be found in Ruddick et al. (2009).

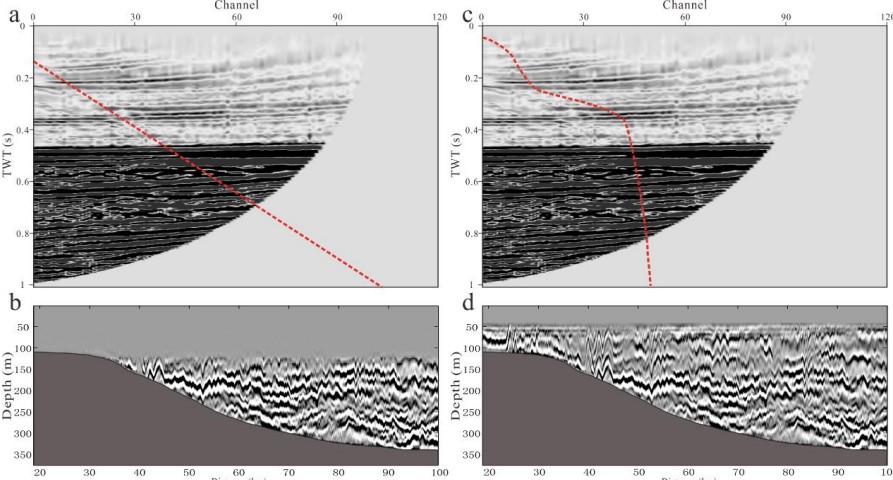

Figure 2. Cutting off the stretch of NMO with a linear function (a) and the corresponding seismic section
(b). Cutting off the stretch of NMO with a custom function (c) and the corresponding seismic section (d).
The red dotted line shows the cut off trace, the right part of seismic data is cut off. The unit TWT of (a)
and (b) is the two-way travel time of seismic wave from source to receiver.

**2.2. Diapycnal diffusivity estimates from seismic data**

Klymak and Moum (2007b) found that the horizontal wavenumber spectrum of the vertical
isopycnal displacement can be interpreted as the internal wave spectrum at low wavenumbers and
the turbulence spectrum at high wavenumbers. The high wavenumber components of spectrum are
dominated by turbulence, and the spectral energy follows the -5/3 power of the wavenumber. The
turbulence part of the horizontal wavenumber spectrum can be expressed by a simplified Batchelor
model (Equation 2-1), so the turbulence dissipation $\varepsilon$ can be estimated from the observed
horizontal wavenumber spectrum. And diapycnal diffusivity can be calculated from Equation 2-2
(Osborn, 1980).


$$\phi_{\varsigma}^{T} = \frac{4\pi\Gamma}{N^2} C_T \varepsilon^{\frac{2}{3}} (2\pi k_x)^{-\frac{5}{3}}$$  (2-1)
$$K_\rho = \Gamma\varepsilon / N^2$$  (2-2)
Where $\phi_{\varsigma}^{T}$ represents horizontal wavenumber spectrum, $\Gamma = 0.2$ is the mixing coefficient, $N$
is the buoyancy frequency, $C_T = 0.4$ is the Kolmogorov constant, $\varepsilon$ represents the turbulence
dissipation, $k_x$ is the horizontal wavenumber, and $K_\rho$ represents the diapycnal diffusivity.

Observations (Nandi et al., 2004; Nakamura et al., 2006; Sallarès et al., 2009) and simulations
(Holbrook et al., 2013) show that the reflection seismic events and isopycnal are spatially consistent.
Therefore, the horizontal wavenumber spectrum calculated from the vertical displacement of the
reflection seismic events is equivalent to the horizontal slope spectrum that Klymak and Moum
(2007b) calculated from horizontal tow measurements. The turbulence dissipation and diapycnal
diffusivity can also be calculated from seismic data (Sheen et al., 2009; Holbrook et al., 2013). First,
we use the seismic interpretation software to pick up reflection events in the seismic section (Figure
3a). Then we calculate the vertical displacement of the reflection events. The vertical displacement
is the distance of the reflection evens deviate from the equilibrium position in the vertical direction.
We take the mean water depth of the reflection events as the equilibrium position. Note that the
choice of equilibrium position will not affect the calculation result. The spectral energy $\phi_{\varsigma}^{T}$ of the
vertical displacement in the horizontal wavenumber domain can be obtained by Fourier transform.
In practical applications, we use the slope spectrum $\phi_{\varsigma_x}^{T}$ instead of the displacement spectrum $\phi_{\varsigma}^{T}$
to distinguish the turbulence subrange from the internal wave subrange. The spectral slope is as
follows (Holbrook et al., 2013):
$$\phi_{\varsigma_x}^{T} = (2\pi k_x)^2 \phi_{\varsigma}^{T}$$  (2-3)
This conversion changes the wavenumber power law in the turbulence subrange from -5/3 to 1/3,
so that it can be distinguished from the internal wave subrange with -1/2 power law (-5/2 in the
displacement spectrum). In calculating the turbulence dissipation in the seismic section, it is
necessary to grid the section and calculate the dissipation in each grid separately. The horizontal
grid is set as 5 km, and the grid step 2.5 km. As the water depth in the seismic data is shallow, the
reflection seismic events are less in the vertical direction. In order to ensure more than two events
in each grid, we set the vertical grid to be 75 m and the grid step 37.5 m. In each grid, we calculated
the spectral slope of each event and took the average as $\overline{\phi_{\varsigma_x}^{T}}$. We fitted the averaged spectrum in
the turbulence subrange to the Equation 2-1 and calculated the turbulence dissipation $\varepsilon$. To reduce
uncertainty, we only calculated the cases with a length >1000 m in each grid. Experiments showed
that this length can correctly represent the slope of energy spectra in turbulence subrange (Figure
3b). After traversing all the grids, the turbulent dissipation section is obtained, and the diapycnal
diffusivity section can be obtained as well according to Equation 2-2. The uncertainty of the


turbulence dissipation and diapycnal diffusivity was evaluated by the error between observed
average slope spectrum and the fitted Batchelor model. We used a spline smoothing function to
smooth the meshing results.

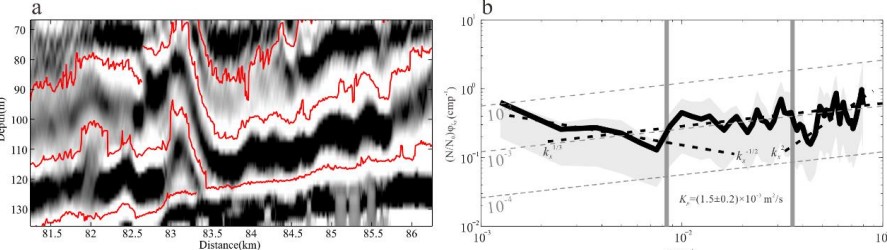

Figure 3. (a) The reflection seismic events in a grid. (b) The average horizontal slope spectrum (black
line). The gray shadow represents the 95% confidence interval. The gray dashed lines represent the
diffusivity contour. The black dashed lines represent the spectral slopes in internal wave subrange,
turbulence subrange and noise subrange, respectively. The gray vertical lines indicate the boundaries of
turbulence subrange.

**2.3. Estimating the horizontal wave-induced velocity of internal solitary wave**


We estimated the wave-induced horizontal velocity of internal solitary wave according to the
method proposed by Moum et al. (2007). This method requires observation data to satisfy two
assumptions: 1) the isopycnal is parallel to the streamline; 2) the internal solitary wave satisfies the
KdV equation. Moum et al. (2007) picked the isopycnal from the high-frequency acoustic section
and fitted it with the KdV equation. The displacement equation of isopycnal can be obtained, and
the derivation of displacement equation is the wave-induced velocity. Seismic data satisfy the first
assumption. Although breaking induced polarity reversal of internal solitary waves close the
streamlines, it is difficult to record reflection seismic data from those areas with closed streamline
at the resolution scale of seismic data. The regional density gradient recorded by the reflection
events still exists, and the streamline is parallel to the isopycnal at this time. While areas with closed
streamlines are strongly mixed, and the density gradient weakens or even disappears, which cannot
be recorded in seismic data. Unfortunately, the internal solitary waves we observed do not satisfy
the second assumption. The KdV equation can simulate internal solitary waves with small amplitude
and weak nonlinearity, but the polarity reversal of the large-amplitude internal solitary waves we
observed cannot be simulated well. Here we did not use theoretical models to fit observations.
Although there are studies using theory to successfully simulate the polarity reversal of internal
solitary waves (Liu et al., 1998; Zhao et al., 2003;), it is difficult to match theories and observations.
We used the picked reflection seismic events to calculate the isopycnal displacement $\eta(x, z)$
(Figure 4b). The $\eta(x, z)$ is the distance that reflection seismic events deviate from the equilibrium
position, which is determined by the mean depth of two shoulders of one internal solitary wave
(Figure 4a). We smoothed $\eta(x, z)$ with a spline function same as that was used for smoothing



turbulence dissipation, so that the resolution of wave-induced velocity is consistent with that of
turbulence dissipation. Therefore, the stream function can be expressed as (Holloway et al., 1999):
$\Psi(x, z) = c\eta(x, z)$            (2-4)
where $c$ is the phase velocity of internal solitary waves. The $c$ can be estimated from pre-stack
seismic data (Tang et al., 2014, 2015; Fan et al., 2021). The seismic data is redundant, because we
have made multiple observations of the same events, which allows us to study the movement of the
water column. Specifically, after sorting the seismic data into CMPs (section 2.1), we extract traces
with the same offset from CMPs to form common offset gathers (COGs). Multiple COGs can be
obtained in the order of offset from small to large. The larger the offset, the lower the signal-to-
noise ratio of the data. We selected the first five COGs to ensure the imaging quality. Pre-stack
migration of COGs yields COG sections. These COG pre-stack migration sections show images of
the same water column at different times. Tracking the change of shot-receiver pairs at a certain
reflection point yields the phase velocity (Fan et a., 2021). Figure 4c shows the change of the shot-
receiver pairs of internal solitary wave trough in the L1 survey line. The straight line represents the
fitting line of the shot-receiver pairs. The average phase velocity of the internal solitary wave during
the imaging time is $c = \dfrac{d_{cmp}}{dt_s}k$, where $d_{cmp}$ is the half of the trace interval, $dt_s$ the time
interval of shot, and $k$ the slope of the fitted line. After calculating the flow function according to
Equations 2-4, the wave-induced horizontal velocity can be expressed as:
$u(x, z) = \dfrac{\partial \Psi}{\partial z}$            (2-5)



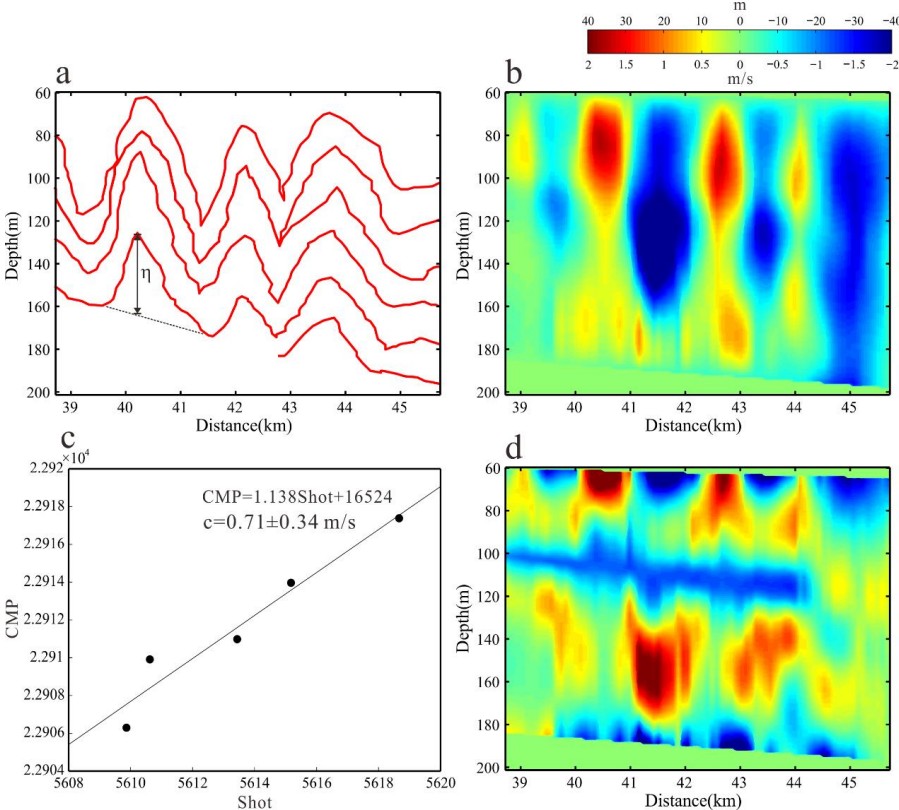

Figure 4. (a) Schematic of calculating internal solitary wave isopycnal displacement using reflection seismic events. (b) The isopycnal displacement section of internal solitary wave. (c) Calculating the mean phase velocity of internal solitary wave by pre-stack seismic data. (d) The wave-induced horizontal velocity.

Note that the wave-induced velocity here is in the seismic resolution scale, which can be regarded as its low-frequency component only. The results are insufficient to characterize the high-frequency components. But such rough wave-induced velocity is useful, because the purpose of calculating wave-induced velocity is for the vertical mixing scheme. The wave-induced velocity makes the resolution scale of the mixing scheme equal to that of mixing parameters estimated from the seismic data, and the two are comparable. In addition, the error of the wave-induced velocity is mainly determined by the error of the phase velocity of the internal solitary wave. For internal solitary waves with polarity reversal, the error of the phase velocity is large, because the phase velocity gradually decreases when the internal solitary wave is shoaling (Bourgault et al., 2007; Shroyer et al., 2008). It can be seen from Figure 4c that the shot-receiver pairs do not completely fall on the fitted line.

**2.4. Mixing scheme for internal solitary waves shoaling**



Shoaling and breaking of internal solitary waves on the continental shelf and slope enhance mixing.
Vlasenko and Huntter (2002) studied the breaking of internal solitary waves over slope-shelf
topography by numerical simulation. In their model, the mixing scheme (PP scheme) proposed by
Pacanowski and Philander (1981) was improved, and a vertical mixing scheme for resolving
breaking internal solitary waves was given. In this scheme, the vertical turbulence kinematic
viscosity and diffusivity are determined by the Richardson-number-dependent turbulence
parameterizations. The expression is as follows:
$$Ri = \frac{N^2}{u_z^2} \qquad (2\text{-}6)$$
$$\nu = \frac{\nu_0}{(1+\alpha Ri)^n} + \nu_b \qquad (2\text{-}7)$$
$$\kappa = \frac{\nu_0}{(1+\alpha Ri)^n} + \kappa_b \qquad (2\text{-}8)$$
Where $u_z$ is the vertical gradient of horizontal wave-induced velocity, $\nu$ is vertical turbulence
kinematic viscosity, $\kappa$ is vertical turbulence kinematic diffusivity. Vlasenko and Huntter (2002)
selected the best model parameters after a series of experiments. They are $\nu_0 = 10^{-3} m^2 s^{-1}$,
$\nu_b = 10^{-5} m^2 s^{-1}$, $\kappa_b = 10^{-6} m^2 s^{-1}$, $\alpha = 5$ and $n = 1$. Based on this model, they simulated the
process of internal solitary wave shoaling and breaking on slope-shelf topography and studied the
breaking criterion.

**3. Results**
**3.1. Polarity reversal of internal solitary wave in seismic section**

When one internal solitary wave propagates cross the transition point, it converts from a depression
wave to an elevation wave. In the two-layer ocean model, the transition point is defined as the
position where the pycnocline is close to the mid-depth (Grimshaw et al., 2010). The three seismic
sections in Figure 5 capture the images of internal solitary waves passing the transition point. Figure
5a is the seismic section of survey line L1. It shows that the water depth becomes shallower from
southeast to northwest, and the bottom slope is steeper between 30-60 km. In the deep-water region
of 60-100 km, internal waves are developed, and the reflection seismic events fluctuate obviously.
Near the seafloor around 80 km, the reflection seismic events are uplifted and discontinuous,
forming a fuzzy reflection area. A mode-1 depression internal solitary wave can be identified at 53
km, indicating that the transition point has not been reached yet. The internal solitary wave has
reversed polarity in the area of 40 km, and a packet of three elevation waves is formed during the
polarity reversal process. The reflection seismic events is continuous here, and no obvious wave
breaking is found. Five elevation waves can be identified around 24-37 km, among which four
elevation waves at 24 km may be formed continuously, while the elevation wave at 37 km formed
later.

Figure 5b gives another internal solitary wave polarity reversal process captured by the survey line
L2. There are two obvious depression waves at 16 km. There are multiple waves with smaller
amplitude around 10-15 km. The polarity of internal solitary wave is reversing within 4-8 km. The
length of the head wave becomes wider and the slope becomes gentler. The leading wave is followed
by a packet of   multiple elevation waves. The reflection seismic events are continuous in the whole
section.

L3 is a cross line whose observation direction is perpendicular to survey line L1 and L2 (Figure 5c).
There are multiple depression waves with large amplitude around 20-35 km, and the reflection
seismic events are continuous. The wave polarity is reversing within 10-20 km, and the reflection
seismic events are discontinuous in this region. At 10 km, there is a large-amplitude elevation
internal solitary wave, and the wave front is almost parallel to seafloor. There is a large-amplitude
depression wave at 17 km, and the wave trough has interacted with topography. The reflection
seismic events before 10 km are discontinuous, and the events near the seafloor are fuzzy. It
indicates that internal solitary waves break and induce strong mixing during the process.

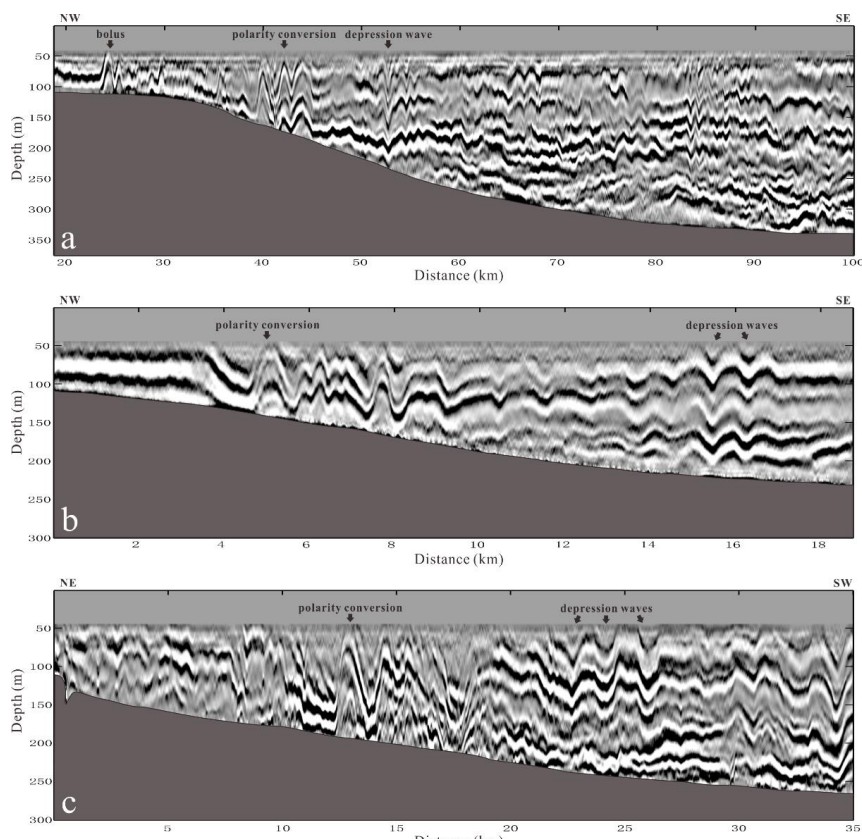


Figure 5. The seismic sections of survey line L1 (a), L2 (b) and L3 (c). The gray regions in the sections
represent seafloor. Internal solitary waves can be clearly seen in all three cases.






**3.2. The horizontal slope spectrum**

We picked the reflection seismic events in the three sections (Figure 7) and calculated the horizontal
slope spectrum using the method described in section 2.2. Figure 6 shows the average horizontal
slope spectrum of the three sections. We calculated the horizontal slope spectrum of all tracked
events and averaged in logarithmic space to determine the wavenumber of turbulence subrange. The
turbulence subrange of the survey line L1 section is 0.005-0.069 $m^{-1}$, as shown by the gray vertical
line in Figure 6a. The corresponding wavelength is 15-200 m. The average diapycnal diffusivity is
$(7.0\pm1.2)\times10^{-4}$ $m^2$/s, which is one order of magnitude larger than the open-ocean value ($10^{-5}$ $m^2$/s).
The spectral energy in internal wave subrange is larger than that in turbulence subrange, indicating
that the energy is dominated by internal waves. This is confirmed by internal waves in the seismic
sections. The difference from Holbrook et al. (2013) is that the calculated horizontal slope spectrum
does not include harmonic noise. This may be because the harmonic noise has been removed when
we filtered the swell noise. In addition, we have not smoothed the events, so some high wavenumber
ranges are reserved. If the events are smoothed, the spectral energy will decrease rapidly in the high
wavenumber range (Holbrook et al., 2013; Tang et al., 2019).

The horizontal slope spectrum of the L2 section is shown in Figure 6b. The turbulence subrange is
0.008-0.068 $m^{-1}$, and the corresponding wavelength is 15-133 m. Compared with the survey line L1,
the turbulence shifts to a smaller scale. The spectral energy in internal wave subrange has the same
order of magnitude as the spectral energy in turbulence subrange, which indicates that the energy is
transferring to small-scale turbulence. This process is closely related to the polarity reversal of
internal solitary waves. The average diapycnal diffusivity is $(1.5\pm0.1)\times10^{-3}$ $m^2$/s, which is two
orders of magnitude larger than the background value.

Figure 6c is the horizontal slope spectrum of the L3 section. It can be seen from the spectrum that
the turbulence subrange is small, ranging from 0.011-0.07 $m^{-1}$. The corresponding wavelength scale
is 14-89 m. The internal wave energy is larger and occupies a larger scale range. It can be seen from
the seismic section of survey line L3 (Figure 5c) that the amplitudes of the internal waves are large.
It indicates that the internal waves carry more energy, so the spectral energy in internal wave
subrange is larger (Figure 6c). In addition, there are many discontinuous and weak reflections in the
seismic section caused by breaking internal solitary waves. Internal solitary wave breaking weakens
the density gradient and enhances local mixing. This phenomenon is most obvious in the survey line
L3, where the average diffusivity is the largest of the three sections, $(2.2\pm0.2)\times10^{-3}$ $m^2$/s.

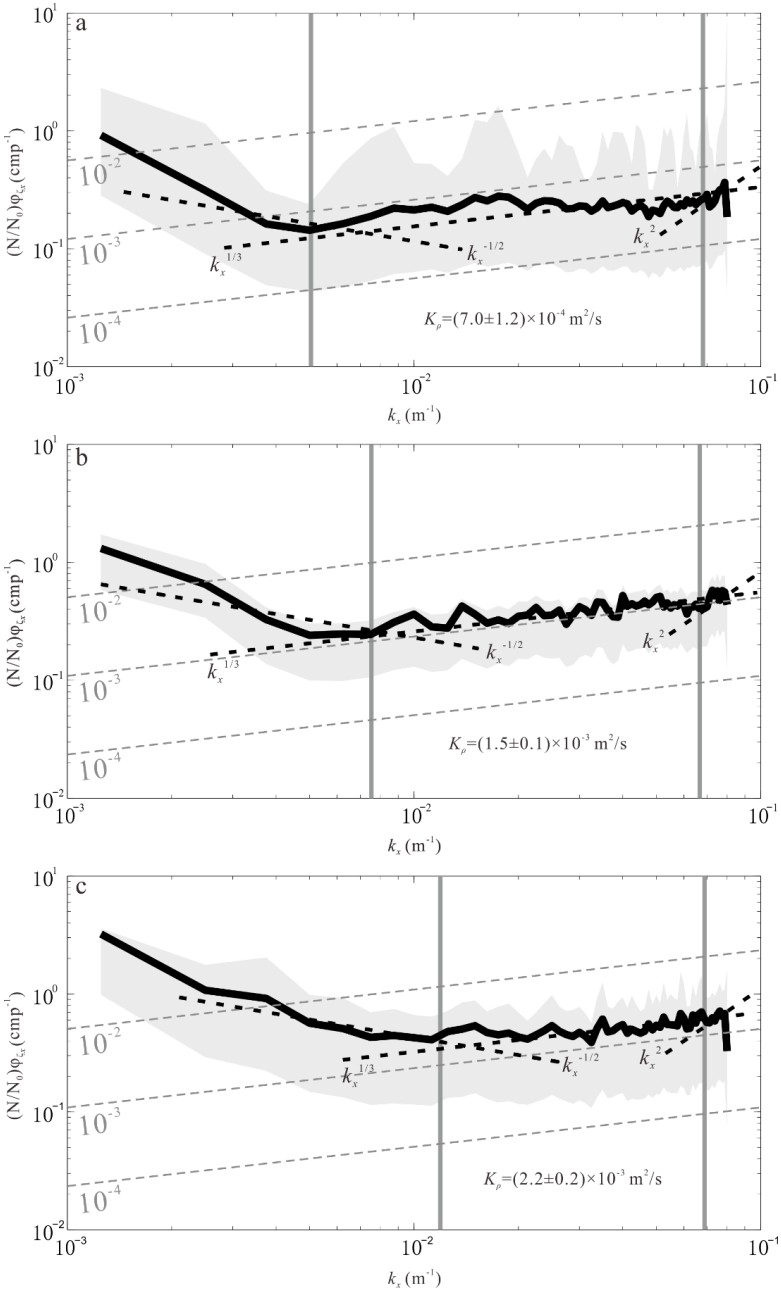


Figure 6. The average horizontal slope spectrum of L1 section (a), L2 section (b) and L3 section (c). The
black line is the spectrum, the gray shadow represents the 95% confidence interval, the gray dashed lines
represent the diffusivity contour, the black dashed lines represent the spectral slopes in internal wave
subrange, turbulence subrange and noise subrange, respectively. The gray vertical lines represent the
boundaries of turbulence subrange.






Figure 6 shows that the spectral energy of the L1 section is smaller than that of the other two sections.
This may be because the imaging range of the L1 section is different. The observations in the L2
and L3 sections are the polarity reversal of internal solitary waves, while the L1 section includes
not only the polarity reversal process, but also internal waves in deep water. The spectral energies
of these two processes should be different. We calculated the average horizontal slope spectrum of
the polarity reversal region and the non-polarity reversal region, respectively (Figure 7). The
spectral energy of the polarity reversal region in L1 section is higher than that of the non-polarity
reversal region, so does diapycnal diffusivity (Figure 7a). It implies that the wave energy will
accelerate to dissipate and transfer to turbulence when its polarity is reversed. Compared with the
non-polarity reversal region, the turbulence subrange of the polarity reversal region is smaller. The
lower boundary of the turbulence subrange of the polarity reversal region is slightly larger than that
of the non-polarity reversal region. It indicates that the turbulence in this region has a smaller scale.
The diapycnal diffusivity in the polarity reversal region in L2 section is about 3 times that of the
non-polarity reversal region (Figure 7b). The turbulence subrange of the polarity reversal region in
L2 section is slightly larger than that of the non-polarity reversal region. From the L2 section, it can
be seen that the events are continuous during the polarity reversal process, which indicates that the
wave breaking is weak. The internal solitary wave gradually fissions into several tails during the
polarity reversal, and energy is dissipated constantly. Therefore, there will be a large turbulence
subrange in the lateral direction (Figure 7b). This process can dissipate much more energy compared
with direct breaking of internal solitary waves (Masunaga et al., 2019). The diapycnal diffusivity in
the polarity reversal region in L3 section is larger, more than 3 times that of the non-polarity reversal
region. Although there are more internal (solitary) waves with larger amplitude in the non-polarity
reversal region, the diapycnal diffusivity is lower. The polarity reversal of internal solitary waves
significantly increases the diapycnal diffusivity. The turbulence subrange of the polarity reversal
region is small, and the lower boundary of the turbulence subrange is greater than 0.01 m$^{-1}$.

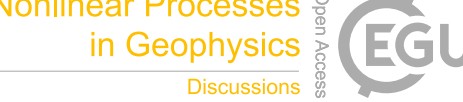
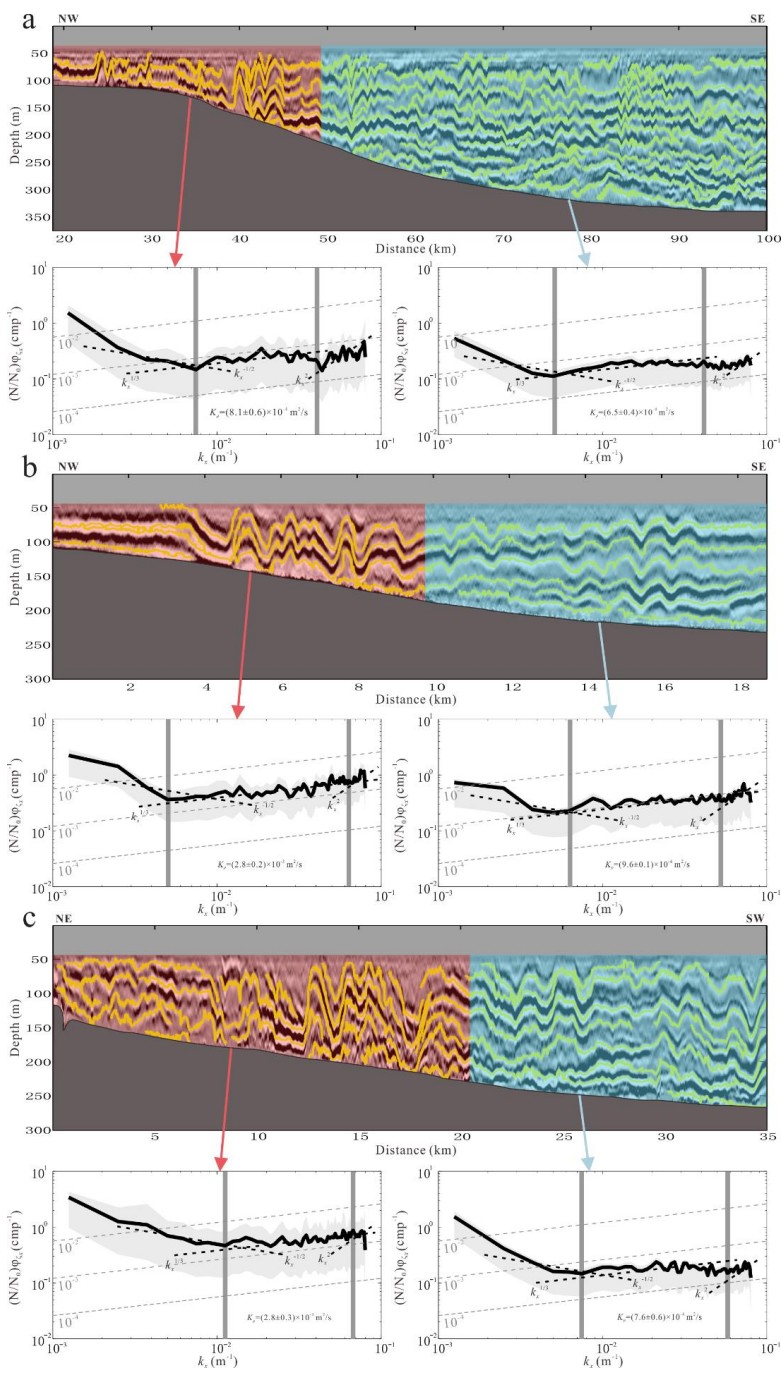

Figure 7. The horizontal slope spectra of the polarity reversal and non-polarity reversal regions calculated from L1 section (a), L2 section (b) and L3 section. The yellow lines are tracked reflection seismic events.



### 3.3. Diapycnal diffusivity maps

The diapycnal diffusivity maps of the three survey lines are shown in Figure 8. Figure 8a shows the map of the survey line L1. The diffusivity is higher than that of the open ocean. The high value presents a patchy distribution, mainly distributed in the depth between 50-150 m. The low diffusivity values are mainly distributed in the depth between 150-300 m. Some high values are also distributed near the seafloor. The diffusivity is larger in the polarity reversal region (24-45 km). Compared the diffusivity of the four adjacent elevation internal solitary waves (24-30 km), we find that the diffusivity is proportional to the amplitude of internal solitary waves. It means that the large amplitude internal solitary waves contribute more to mixing. In the polarity reversal region (40-45 km), the diffusivity of the head wave's front is higher, that is, where the slope of the wave front becomes gentle. While the diffusivity of the two elevation waves followed the head wave is small. It indicates that the mixing induced by the internal solitary wave polarity reversal is stronger at the beginning, and more energy is dissipated at this time. In the non-polarity reversal region (50-100 km), the diffusivity is low. The mode-1 depression internal solitary wave at 52 km increases the diffusivity. There is an abnormal reflection area near the seafloor at 80 km, and the diffusivity is high. In addition, there is also an area with increased diffusivity between 100-250 m at 93 km. This may be related to the activity of large amplitude internal waves.

The diffusivity map of the survey line L2 is shown in Figure 8b. The high value is mainly distributed at the front of head wave during polarity reversal process (4 km), which is consistent with the characteristics on L1. The diffusivity after the head wave is low, but it is still higher than that in other regions. The diffusivity in the non-polarity reversal region is almost uniform. The two internal solitary waves at 15-16 km did not increase the diffusivity. There is a low diffusivity area near the seafloor around 16-18 km, which is caused by not tracked reflection seismic events in this area. The diffusivity map of survey line L3 (Figure 8c) is similar to that of L2. The high value is distributed in the polarity reversal region and the diffusivity of head wave is still high. However, unlike the diffusivity map of L2, the high diffusivity is mainly distributed in the shallow part of the head wave (water depth 50-120 m), while the diffusivity of the whole head wave in L2 is high. In the non-polarity reversal region, the diffusivity is small and the distribution is uniform too. The diffusivity near the seafloor at 25-27 km is slightly lower than other regions.



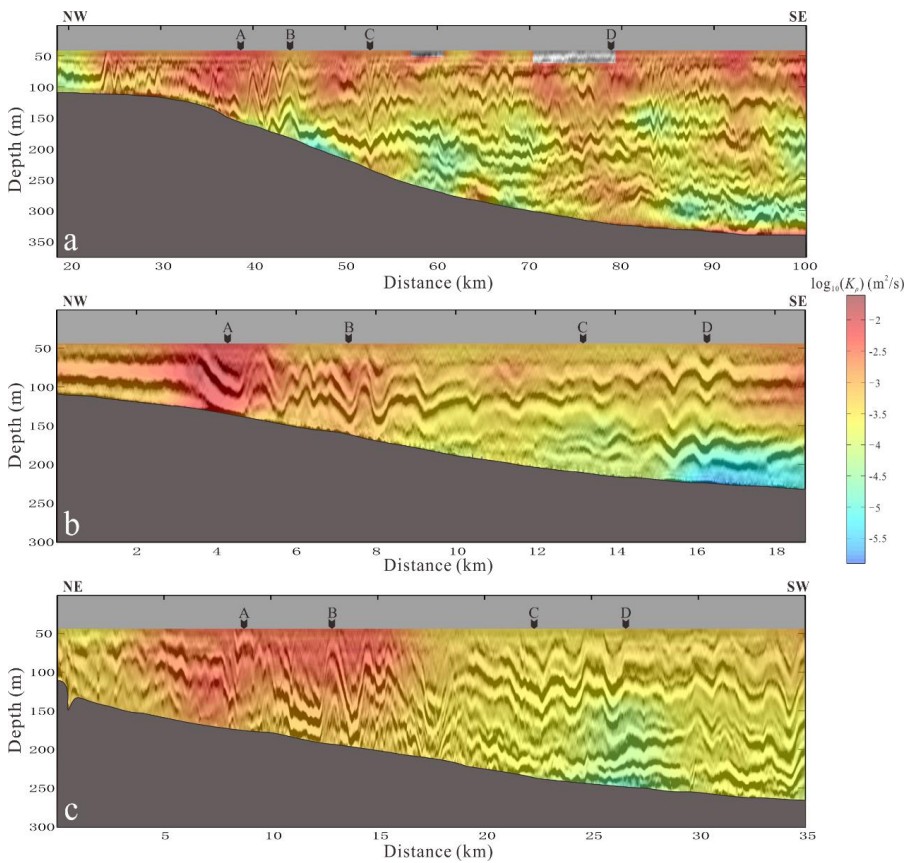

Figure 8. The diapycnal diffusivity map of survey line L1 (a), L2 (b) and L3 (c). The black arrows represent the position of vertical diffusivity profile.

## 4. Discussions

### 4.1 The relationship between diffusivity and reflection seismic events

When there is a significant impedance difference in the water column, a reflection seismic event will occur (Holbrook et al., 2003; Ruddick et al., 2009). The impedance difference in the ocean is contributed by temperature gradient and salinity gradient, where the former is usually greater than the latter (Ruddick et al., 2009; Sallarès et al., 2009). Density is a function of temperature and salinity, so the reflection seismic events are related to the density gradient. The enhanced mixing reflects the structure of density gradient, thereby changing the appearance of the reflection seismic events. Understanding the relation between diffusivity and reflection seismic events can help us analyze the spatial distribution of diapycnal mixing by seismic section. Figure 8 shows that the reflection seismic event in the high diffusivity region is obviously different from that in the low diffusivity region. In the high diffusivity area (red in Figure 8), the reflection seismic events are fuzzy, discontinuous or bifurcate. While in the low diffusivity area (yellow and blue in Figure 8), the reflection seismic events are clear and continuous. This is because regions with high diffusivity




are strongly mixed. The density gradient is changed by mixing, so that it affects the appearance of
reflection seismic events. For example, in the polarity reversal region of three seismic sections, the
diffusivity is high, and the reflection seismic events are fuzzy and discontinuous. Especially in the
range of 5-10 km in Fig. 8c, the events are obviously break and weak. The diffusivity is low in areas
where the events are clear, such as the region near the seafloor around 45-50 km and the region near
the sea floor around 93-99 km in Figure 8a, and the region near the sea floor around 24-27 km in
Figure 8c.
The diffusivity is not only related to the continuity of the reflection events, but also related to the
fluctuation intensity of the events. The greater the fluctuation intensity of the events, the higher the
spectral energy, and the greater the diffusivity value. There is a mode-1 depression internal solitary
wave at 50-58 km in Figure 8a, and the reflection seismic event is clear and continuous at 180 m.
But the diffusivity is high, because the reflection events fluctuate more strongly. It can be seen from
the figure that, in addition to the amplitude of the internal solitary wave, there are also many high-
frequency waves at the shoulders of the internal solitary wave. These waves increase the spectral
energy and result in a higher diffusivity. In addition, the reflection seismic events before 4 km in
Figure 8b is continuous and without obvious fluctuations, but the diffusivity is higher. It can be seen
from the figure that the reflection events of this region are thicker than that of other regions. The
seismic data processing of the three sections in Figure 8 is the same, so the thicker events in Figure
8b do not stem from the low frequency of seismic waves. We think this may be caused by small-
scale mixing between layers, such as K-H instability. Figure 9 is an enlarged view of the location
between 2-3 km in the seismic section of L2 (Figure 5b). The wavelength of the seismic wave (red
line) at 80 m is larger than that at the seafloor, which is formed by the overlap of multiple wavelets.
It can be seen from the figure that a weak reflection event is barely visible at 80 m, which indicates
a thin reflection layer with weak impedance differences. The K-H instability can continue for a long
distance in the lateral direction (Seim and Gregg, 1994; Haren et al., 2014; Chang et al., 2016; Tu
et al., 2019) and enhance local ocean mixing. This structure can form at the tail of internal solitary
wave (Moum et al., 2003). The vertical scale of the K-H instability is small and usually appears on
the isopycnal. On the one hand, K-H instability weakens the density gradient so that the reflected
seismic wave energy is reduced. On the other hand, the vertical scale of K-H instability is lower
than the seismic wave resolution (a quarter of the seismic wave wavelength), so it makes overlapped
wavelets and stretched wavelength (Figure 9). Therefore, the reflection event in this area is thicker.
Besides, the horizontal scale of the K-H instability train is large, which may explain the larger
turbulence subrange on the horizontal slope spectrum (Figure 7b).

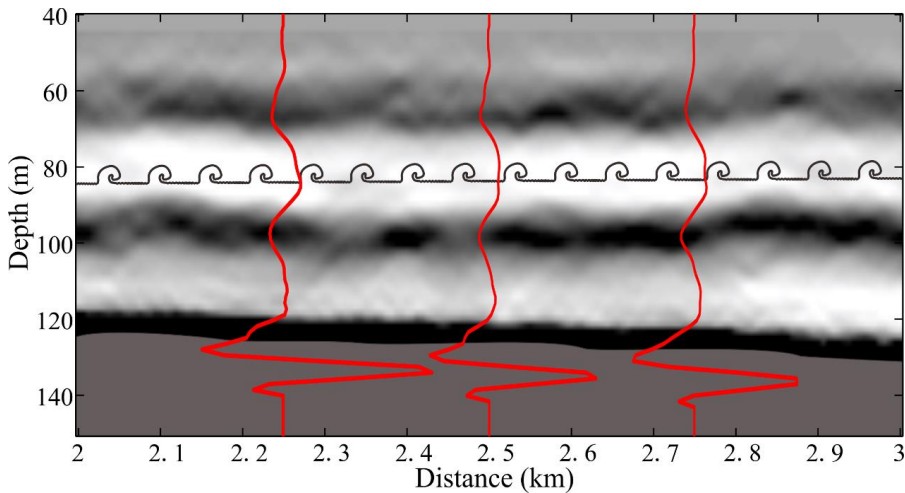


Figure 9. Schematic of the K-H instability. The red lines are the seismic waves, and the black billows
represent the K-H instability.

**4.2 Enhanced diapycnal mixing induced by the polarity reversal of internal solitary waves**

Strong mixing in the ocean mainly occurs near rough topography or area with strong tides (Simpson
et al., 1996; Rippeth et al., 2001, 2003; Nash and Moum, 2001; Klymak et al., 2008; Jarosz et al.,
2013; Staalstrøm et al., 2015; Wijesekera et al., 2020; Voet et al., 2020). The Dongsha Atoll region
in the South China Sea possesses both above features. On one hand, the Dongsha Atoll lies on the
continental slope with variable topography. On the other hand, large amplitude internal solitary
waves (Alford et al., 2015) propagating from the Luzon Strait reflect, refract, and shoal in this region.
This process will dissipate most of the energy carrying by the internal solitary waves. Especially in
the shoaling process, polarity reversal and breaking occur and the energy of internal solitary waves
transfer to smaller-scale waves. Our results (Figure 6) indicate that the average diffusivity has the
magnitude order of O(-4)-O(-3), consistent with previous observations by other techniques. St.
Laurent (2008) observed turbulent mixing on the continental shelf and slope, and found that the
mixing is higher at the shelf break, and the magnitude order of average dissipation is O(-7)-O(-6).
According to the average buoyancy frequency $N = 6cph$, the magnitude order of the average
diffusivity is O(-4)-O(-3) and consistent with our result. Yang et al. (2014) observed diapycnal
mixing on the continental shelf and slope, and found that the average diffusivity can reach the
magnitude order of O(-3) too. Similar results have been reported in the study of internal solitary
waves shoaling in other regions. For example, Sandstrom et al. (1989) observed the turbulent
diffusivity caused by the nonlinear internal wave group on the continental slope of Canada, and
found the average diffusivity of $2.4 \times 10^{-3}$ m$^2$/s. Carter et al. (2005) observed the elevation internal
solitary waves in Monterey Bay and a diffusivity on the magnitude order of O(-4). Richards et al.
(2013) observed the shoaling of nonlinear internal waves at the St. Lawrence Estuary, which induced
high turbulence and enhanced mixing. Therefore, it is reasonable that diapycnal mixing induced by
nonlinear internal waves on the continental shelf and slope in the northern South China Sea can





reach 100 times that in the open ocean.

The high diffusivity is mainly in the leading internal solitary wave during the polarity reversal. We
suggest that strong mixing may be caused by internal wave breaking due to convective instability.
In Figures 8a and 8c, the reflection seismic events are obviously discontinuous in the high
turbulence area of the leading wave, indicating that the density gradient is weakened by internal
wave breaking. The trough of the internal solitary wave decelerates first when the polarity is
reversed (Shroyer et al., 2008), which makes the Froude number (Fr) greater than 1 and causes
convective instability. This phenomenon can be found in other observational data. In the high-
frequency acoustic section, the backscatter at the top of internal solitary wave is increased when it
changes from depression to elevation wave (Orr and Mignerey, 2003), which indicates that the
turbulence of the front increased. However, in the seismic section of Figure 8b, we did not find the
events break at the front the polarity reversal internal solitary wave. The strong mixing of this
internal solitary wave may be induced by shear instability (Figure 9). Therefore, both convective
instability and shear instability are responsible for the enhanced mixing in this process. In addition,
the non-polarity reversal region in Figure 8a has a higher diffusivity in 50-150 m than other regions.
This range is in the thermocline (Figure 1c). The internal waves usually greatly increase mixing in
the thermocline, which is related to the shear instability of internal waves (Mackinnon and Gregg,
2003). Shear instability is an important mechanism of internal wave dissipation (Farmer and Smith,
1978), and it more likely occurs in nonlinear internal waves than convective instability (Zhang and
Alford, 2014). The results of high-frequency acoustic observations show that the enhanced
backscatter at the bottom of the thermocline represents higher shear instability when the internal
solitary waves are shoaling (Orr and Mignerey, 2003), which is consistent with the depth range of
high diffusivity in our results.

What is inconsistent with the observed distribution of mixing is that our results are not able to show
diffusivity in the bottom boundary layer. Because our seismic data was collected in summer, the
strong stratification at this time limits the vertical range of the bottom boundary layer (Mackinnon
and Gregg, 2003). So that the bottom boundary layer near the Dongsha Atoll is thin and lower than
the thickness that can be recorded by seismic data. So, the diffusivity we calculated does not include
the bottom boundary layer. The enhanced diapycnal mixing induced by the polarity reversal of
internal solitary waves plays an important role in local environment and primary productivity. On
one hand, diapycnal mixing on the continental slope and shelf makes an important contribution to
ocean heat flux, which affects climate and the ocean through heat exchange of local water column
(Rahmstorf, 2003; Tian et al., 2009). On the other hand, the vertical flux caused by turbulence can
redistribute materials in the ocean and have an important impact on the marine ecological
environment (Sharples et al., 2001; Moum et al., 2003; Klymak and Moum, 2003; Wang et al., 2007).

**4.3 The mixing scheme of internal solitary wave shoaling**

We compared the vertical distribution of diffusivity with the vertical mixing scheme of internal
wave breaking proposed by Vlasenko and Huntter (2002). Although Klymak and Legg (2010) also
proposed a mixing scheme for internal wave shoaling and achieved good results in numerical
simulation, we cannot use this method to calculate mixing parameters because of lacking high




resolution density observation data. Figure 10 shows the vertical distribution of diffusivity from
seismic data (solid line) and the diffusivity calculated from mixing scheme (dashed line) at 4
positions of the three survey lines (black arrows in Figure 8). The reflection events in the L3 section
are broken, and it cannot be guaranteed that the events are parallel to the streamline. Therefore, we
did not use the method described in section 2.3 to calculate the wave-induced velocity, and thus did
not obtain the diffusivity of the mixing scheme. It can be seen from Figure 10 that the turbulent
diffusivity gradually decreases from shallow to deep water. Except for the local low diffusivity value
in the deep water at the position D of Figure 10b and 10c, the diffusivity reduction rate at the other
location is similar. Figures 10a and 10b show that the parameterized diffusivity is nearly 2--3 orders
of magnitude smaller than our result, but they have a similar trend of change. In Figure 10a (line
L1), the parameterized diffusivity (blue dotted line) at position B decreases an order of magnitude
within 50-100 m. This tendency is same as our results. However, the parameterized diffusivity
within 150-200 m increases by one order of magnitude, which is inconsistent with our results (solid
blue line). The parameterized diffusivity at position C fluctuates periodically and also keeps a
decreasing trend on the whole. In the survey line L2, we selected position A and position B to
calculate the parameterized diffusivity. The diffusivity at position A (red dashed line) decreases
rapidly within 60-100 m, and then almost keeps unchanged. This is different from our result (solid
red line), and the reduction rate of the diffusivity is larger than our result. The trend of the diffusivity
at position B (blue dashed line) above 110 m is consistent with our results (solid blue line), but the
diffusivity below 110 m decreases rapidly and then rises again. In our results, the diffusivity
decreases slowly at the same depth. The value is consistent with that in the open ocean. However,
the mixing enhanced obviously on the continental shelf and slope, because of the internal wave
shoaling. The mixing scheme underestimates mixing, especially the strong mixing induced by the
polarity reversal of internal solitary waves. Our results indicate that near the Dongsha Atoll, where
large amplitude internal solitary waves develop, mixing will be enhanced by the shoaling internal
solitary waves. The diffusivity gradually decreases from shallow to deep water (not including the
bottom boundary layer). This has important implications for improving the mixing scheme for
models on the continental shelf and slope.

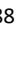

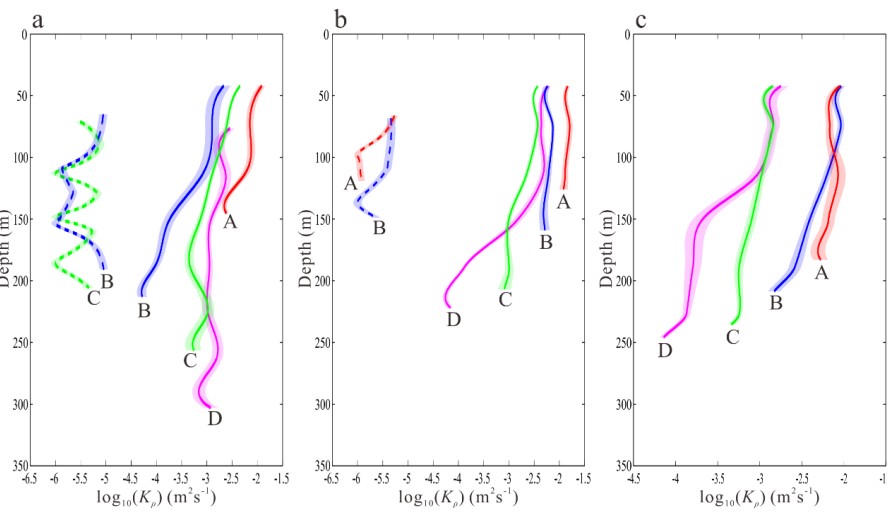






Figure 10. The vertical distribution of diffusivity from seismic data compare with the mixing scheme. The solid line represents the vertical distribution of diffusivity at the four positions A (red), B (blue), C (green) and D (magenta), and the dotted line represents the parameterized diffusivity at the corresponding positions. The shadow indicates the margin of error.

**5. Conclusions**

We have observed the polarity reversal of internal solitary waves by reflection seismic data near the Dongsha Atoll in the South China Sea, and calculated their slope spectra (Figure 6) and diapycnal diffusivity (Figure 8). The results show that the average diapycnal diffusivities of the three survey lines are about two orders of magnitude greater than the open-ocean value. We calculated the average spectral slope of the polarity reversal and non-polarity-reversal regions (Figure 7), and found that the former is about 3 times larger than the latter. The diffusivity maps reveal that horizontally high diffusivity is mainly in the leading wavefront of an internal solitary wave in reversing polarity, and vertically high diffusivity is mainly in the thermocline (50-100 m).

We analyzed the relation between reflection seismic events and diapycnal diffusivity. The result indicates that continuous and clear reflection events correspond to low diffusivity, while discontinuous or fuzzy events correspond to high diffusivity. The strength of the events also affect the magnitude of diffusivity. The stronger the fluctuation, the higher the spectral energy, and the higher the diffusivity. In addition, we observed an area of high diffusivity with a large horizontal scale in L2, and the reflection events did not appear to be discontinuous or fuzzy. We suggest that this is enhanced mixing induced by the K-H instability (Figure 9). The vertical scale of the K-H instability is smaller than the resolution of our seismic data, so we cannot observe clearly in the seismic data. But its high-energy characteristics can be recorded by reflection events.

Our results show that shoaling internal solitary wave enhance local mixing. The magnitude order of diapycnal diffusivity is consistent with previous studies. We suggest that there are two mechanisms accounting for the enhanced mixing. On one hand, the polarity reversal of internal solitary waves results in convection instability, which induces internal solitary wave breaking. This mechanism appears at the leading edge of one internal solitary wave in the survey lines L1 and L3. The discontinuous reflection events indicate that the internal solitary wave is broken. While in the seismic section of L2, the reflection events are continuous and clear at the leading edge of the internal solitary wave and other strong mixing areas in the three sections. Such strong mixing is mainly caused by shear instability.

We picked four positions from the diffusivity maps to analyze the vertical distribution of diapycnal diffusivity (Figure 10).Our result shows that the diffusivity gradually decreased from shallow to deep water (excluding the bottom boundary layer). Compared with previous one mixing scheme, the parameterized diffusivity is about 2--3 orders of magnitude smaller than our result. This means that the mixing scheme underestimates mixing induced by internal solitary wave shoaling near the Dongsha Atoll. However, the vertical pattern of the parameterized diffusivity is consistent with our result.





**Code and data availability.** The bathymetry data were provided by the General Bathymetric Chart of the Oceans (GEBCO, http://www.gebco.net/), and prepared using the Generic Mapping Tools (GMT, https://generic-mapping-tools.org/). The hydrological data set we used were product by Copernicus Marine Environment Monitoring Service (CMEMS, https://resources.marine.copernicus.eu/). The seismic data were processed using Seismic Unix (https://wiki.seismic-unix.org/start/).

**Author contribution.** The concept of this study was developed by Haibin Song and extended upon by all involved. Yi Gong implemented the study and performed the analysis with guidance from Haibin Song. Zhongxiang Zhao, Yongxian Guan, Kun Zhang, Yunyan Kuang and Wenhao Fan collaborated in discussing the results and composing the manuscript.

**Competing interests.** The authors declare that they have no conflict of interest.

**Acknowledgements**. The seismic data own to the Guangzhou Marine Geological Survey (GMGS). Thanks to the GMGS for providing 2D seismic data. This work is supported by the National Natural Science Foundation of China (Grant Number 41976048); and the National Key R&D Program of China (2018YFC0310000).

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
