# Peer review of "Enhanced diapycnal mixing by polarity-reversing internal solitary"

_Nonlinear Processes in Geophysics, 2021_

## Author Comment (AC2)

**Referee #2**

The applied method is also of interest. The general comment is related to using chain assumptions, filtering windows, data from different sources (e.g. stratification parameters from Copernicus), etc. The question is: Would authors provide estimates of the sensitivity of calculated dissipation and diffusivity to the used in calculations parameters/assumptions because of the lack of comparison with dissipation/diffusivity data obtained using other methods?

*Response: Thank you for your time and constructive comments. According to formulas 2-1 and 2-2, the parameters used in calculating the diffusivity are N, $\Gamma$, and $C_T$. It can be seen from the formula that the diffusivity is proportional to N. The mean deviation of the buoyancy frequency we use is about 2% (Figure 1), so the uncertainty of the corresponding diffusion rate is about 0.008 logarithmic units. In addition, the diffusivity is proportional to $\Gamma^{-1/2}$, and the uncertainty of $\Gamma$ is 0.1-0.4, so the corresponding uncertainty of the diffusivity is 0.15 logarithmic units. Similarly, the diffusivity is proportional to $C_T^{-3/2}$. The uncertainty of $C_T$ is 0.3-0.5, and the corresponding diffusivity uncertainty is 0.15.*
*In addition, the key reason for the uncertainty of the diffusivity in our calculation is the fitting of the Batchelor model and the slope spectrum (Figure 3b and Figure 6). We evaluated the uncertainty of the diffusivity based on the fitting error. The fitting error is the least squares standard deviation between the Batchelor model and the slope spectrum. The following figure is the uncertainty of the diffusivity. We have added an Appendix in the manuscript to illustrate the uncertainty of the diffusivity.*

[Figure]

The uncertainty of diffusivity in Figure 8 of the manuscript.

**Minor comment1:** l. 26 " the difference between our and previous diffusivity profiles is about 2-3 orders of magnitude,…". May be better to write "mixing scheme based on Richardson number dependent turbulence parameterizations instead of "previous diffusivity profiles"?

*Response: Thanks for your suggestion, we have changed the statement in the manuscript.*

**Minor comment2:** l.64 O(-3) m2 s-1 Please, use standard order designation O(10^-3) through the paper and correct misprint.

*Response: Thanks for your suggestion, we have corrected the misprint.*

**Minor comment3:** l.274 and l. 284 "Huntter" read as Hutter

*Response: We have corrected this spelling error.*

---

## Author Comment (AC3)

**Referee #1**

Three survey lines of seismic data were used to reveal internal solitary waves and mixing properties. The specific polarity-reversing processes of internal waves were found to induce energetic mixing. The mixing distribution and driving factors are discussed. This paper is very interesting, with some innovative properties on internal wave mixing revealed. So I think this paper quite deserves publication before some comments and concerns being addressed and responded.

*Response: First of all, thank you for your support to our work, we have carefully considered your advises and revised the manuscript.*

**Comment1:** Title, suggest includes the seismic method as used, since this is a very valuable and attractive method used for internal wave research.

*Response: Thanks for your suggestion, we have changed the title to **Enhanced diapycnal mixing with polarity-reversing internal solitary waves revealed by seismic reflection data.***

**Comment2:** Method, The N from the reanalysis data was used to estimate the diffusivity. That means N and other variables in the equations 2-1, 2-2 are at the different time, so as for the mixing scheme evaluation. Should clarify the reasonability of time differences.

*Response: We chose the buoyancy frequency from July to August in 2009 to calculate the average buoyancy frequency, which is the same as the seismic data acquisition time. We added a description in lines 118-120 of the manuscript ('Besides, since the buoyancy frequency changes seasonally, we only selected the buoyancy frequency from July to August in 2009, which matches the seismic data observation time.').*

**Comment3:** One of the main finding is that, mixing intensifies during the polarity reversal process. If this is a general conclusion, I believe some other studies based on numerical simulations and other field data will also support this point. If not, some explanations on the detailed physical process and mechanisms should be added. The general knowledge is that, polarity reverse does not directly correspond to breaking and mixing.

*Response: For weakly nonlinear internal solitary waves, significant breaking and mixing may not occur during the polarity reversal. However, for the large amplitude internal solitary waves in the South China Sea, due to the large energy and nonlinearity, the polarity reversal may be accompanied by breaking and mixing. Vlasenko and Hutter (2002) carried out a numerical simulation of large-amplitude internal solitary wave breaking. The results show that the cumulative effect of nonlinearity leads to the rear of internal solitary wave steepening to form polarity reversal or overturning to form breaking, which is very similar to the structure of the internal solitary waves we have observed. Although the rear of the internal solitary waves in our section do not overturn, local instability is easy to occur when the rear becomes steep. It should be noted that the internal solitary wave breaking mentioned in our article refers to the breaking caused by this local instability,*

*rather than the classic four types of breaking. We have added statements in Line 328-329 ('It should be noted that the breaking mentioned in this article refers to local breaking caused by instability, not the four types of classic four types of breaking (Aghsaee et al., 2010).'). In addition, Orr and Mignerey (2003) observed the polarity reversal of internal solitary waves in the South China Sea, which showed that the increase in the intensity and thickness of the scattering layer is related to local mixing. This is consistent with our point of view. Therefore, we believe that for the large-amplitude internal solitary waves in the South China Sea, the polarity reversal will be accompanied by local breaking to cause mixing enhancement.*

**Comment4:** In Figure 5, as said by the author, occurrence of internal wave breaking, cannot be discerned by the readers. May give an enlarged regional image around 5 Km from Figure 5c.

***Response:*** *In Figure 5c, the internal solitary wave breaking is judged by the continuity of the reflection events. As we discussed in Section 4.1, the shape of the reflection events corresponds to the structure of the density gradient. If the internal solitary wave is not broken, the reflection event is clear and continuous, as shown in the range of 20-30 km in Figure 5c. If the internal solitary wave breaks, the structure of the density gradient will be destroyed and result in the reflection events break. The density gradient is weakened in the area with strong mixing, the reflection events are fuzzy. In Figure 5c, the reflection events are discontinuous and fuzzy in the area before 10 km. We think this may be caused by the internal solitary wave breaking. We have modified the description of lines 325-328 ('Most of the reflection seismic events before 10 km are discontinuous and fuzzy, especially in the range of 6-10 km (Figure 5d). It indicates that the reflective structures in this region may be destroyed by internal solitary wave breaking.') and added a subfigure in Figure 5c to show the image before 10 km.*

[Figure]

*Figure 5. The seismic sections of survey line L1 (a), L2 (b) and L3 (c). The gray regions in the sections represent seafloor. Internal solitary waves can be seen in all three cases. The subfigure (d) is the enlarged regional image of 6-10 km.*

**Comment5:** For the discussions of convective and shear instability, as in Figure 9, the authors should provide clear and direct evidences. In Figure 9, nothing about the instability can be clearly seen. Also, why provide the regional section from Figure 5b, but not from Figure 5c with much more instability and mixing as mentioned.

*Response: In Section 4.1, we discussed the characteristics of reflection events corresponding to high diffusivity. We find that the high diffusivity in Fig. 5b and Fig. 5c corresponds to different reflection events. In the region before 4 km in Figure 5b, the diffusivity is high, and the reflection events are continuous and clear. This is completely different from the discontinuous events with high diffusivity in Figure 5c. Therefore, we separately explained the mechanisms of these two high diffusivities. We believe that the discontinuity of the reflection event indicates that the internal solitary wave is broken, which may be caused by the convective instability when the polarity is reversed. The*

*explanation we give for the high diffusivity in Figure 5b is the K-H instability, and we provided some evidences from seismic data (Figure 9). Since it is easier to accept the view that the discontinuous events corresponds to a high diffusivity, so we separately discuss the situation in Figure 5b. The discussion of convective instability and shear instability is the most likely explanation we have given based on our observational data. Unfortunately, we do not have other data to support our explanation. We quoted the opinions of some other researchers to support our explanation. We have revised our statements in the Section 5 (conclusions) to make them appear less arbitrary.*

**Comment6:** Lines 37-38, only one reference is not enough to indicate the commonality of internal tides and internal waves on the global continental shelves and slopes.

*Response: We have added some references in Line 39-40.*

**Comment7:** Lines 58-59, Internal solitary waves in the northeastern South China Sea, can be generated from the Luzon Strait as the authors told. But most internal solitary waves are generated from the nonlinear steepening of internal tides remotely from the Luzon Strait or locally generated from the continental slope (Min et al., 2019, GAFD; Xu et al., 2016 JGR).

*Response: Thank you for pointing out our mistake, we have revised our statement in Line 59-61.*

**Comment8:** Lines 73-75, some references on the Kuroshio and its connection to internal waves should be added. Like Jan et al., 2011 JGR, Xu et al., 2021 JGR.

*Response: Thanks for your suggestion, we have added a few references in Line 77.*

**Comment9:** Lines 44-45, please confirm that 73% of internal wave energy can be attributed to internal solitary waves.

*Response: Apologize for our ambiguous expression. From simulations Bogucki et al. (1997) have calculated that up to 73% of the energy of the internal wave field can be carried upstream by ISWs. We have modified the expression to '**Numerical simulation results indicate that up to 73% of the internal wave field energy can be carried by internal solitary waves.**'.*

**Comment10:** Line 47, researches-> research.

*Response: We have corrected this spelling error.*